# Jewish Education in Algerian Jewish Communities—Multiple Identities in an Era of Change (1830–1939)

**Yossef Charvit**

Israel & Golda Koschitzky Department of Jewish History and Contemporary Jewry, Bar Ilan University, Ramat Gan 5290002, Israel; yossef.charvit@biu.ac.il

**Abstract:** Our discussion of the Algerian Jewish community's Jewish education during the French period (1830–1939) sheds light on the community's multiplicity of identities viewed through the prism of the Spanish diaspora and French colonialism and sociology. Algerian Jewry's multiple identities during the French period originate in the community's education, both general and Jewish. The Jewish education in Algeria fueled loyalty to one's Jewish identity and heritage and partially knit together the fissures that materialized in Jewish society ever more forcefully in this era of change. This article proposes a new methodological and historiographical approach based upon the examination of diverse sources—from communal and colonial sources to rabbinic, consular, and scholarly ones—using them to present a complete and multidimensional historical picture. Recognizing the many identities adopted by Algerian Jews during the French period is indispensable to conducting balanced and quality research into Algerian Jewry's history. The complexity of Algerian Jews' identity during the French period was the source of its richness and amplitude and a point of contention in the historiographical research of Algerian Jewish history.

**Keywords:** Algerian Jews during the French period; Alliance Israélite Universelle; Jewish education in Algeria; Israelite Central Consistory of France; Adolphe Crémieux

## 1. Introduction

Scholars of Jewish history should be conversant with the Hebrew and general sources of the period being investigated. This observation, made about Antiquity and the Middle Ages, is no less true of the Modern period. However, the abundance of readily available sources related to the Modern period may lead an investigator to adopt mistaken beliefs that divert his or her attention. However, in order to become familiar with the complex dynamics of internal processes of the historical experience and to faithfully represent the historical picture in all its complexity, the historian of the Modern period must also know the Hebrew sources. Relying solely upon general sources leads the scholar to draw a one-sided, one-dimensional, and ultimately distorted, historical picture; analyzing the historical moment from a full host of perspectives is indispensable.

I have endeavored to paint a picture that is faithful to the many archives and scholarly studies dedicated to Algerian Jewish history. However, rather than being satisfied solely with the French and Judeo-French archives popularized by scholars (including the Archives d'Outremer in Aix-en-Provence, the Archives Nationales in Paris, Archives du Ministère des Affaires Etrangères, Quai d'Orsay in Paris, the Archives du Consistoire Central in Paris, and the Archives de l'Alliance Israélite Universelle in Paris), in this study, I generally rely upon a diverse set of archives, including consular ones, as well as unpublished rabbinic sources and Hebrew and scholarly archives (ACHJ, Jerusalem; Archives Sionistes, Jerusalem; Archives diplomatiques, Nantes). Studying these sources leads us to a new form of historiography that I have termed "Hebrew historiography." Only after overcoming the methodological limitations imposed by the Hebrew sources as historical ones can the historian reconstruct the tapestry of Jewish life based upon its own internal logic. The reason is that these texts

are not apologetic but testimony to the substantive—not the ideological—continuum of Jewish life. Thus, these sources more faithfully depict the true historical reality rather than convey an impression or interpretation of it. French historiography—which until now has played the traditional and dominant role in the study of Algerian Jewry—and Hebrew historiography do not merely parallel one another but actually enrich and enhance one another, broadening and deepening the historical picture (Attal 1996; Allouche-Benayoun and Bensimon 1989; Ayoun and Cohen 1982; Chemouilli 1976).

Therefore, in my research, I seek to correct a historiographical lacuna reaching back many years, and to stop looking at Algerian Jewish history from the outside. Thus, the goal here, to paraphrase Professor Shmuel Trigano, is to return the study of the history of Algerian Jewry to the field of Jewish history, for until now, it has been relegated to the field of French history because most of its sources are in French (Thesis defense, 14 January 1999).

Taking this approach, the study of the connection between Algerian Jewry and the Land of Israel, the study of Zionism, the study of the traditional and modern worlds, and the study of the nature of Algerian Jewry's secularization all rest upon intra-community data and communal trends which provide a picture unlike the prevailing scholarly consensus. For example, a study of Algerian Jewish education that relies solely upon the data stored in the Alliance Israélite Universelle communal archives is not undertaking its task faithfully. This is because it overlooks the religious and pedagogical materials produced by the local rabbis who molded Jewish education and who were entrusted with addressing the complexities facing their community's members. These included the colonial heritage of secularization—a secularization unique to the Islamic lands, in general, and to Algeria, in particular.

Traditional research into Algerian Jewry ultimately finds itself mired in paradox and ambiguity. In this work, I have tried to resolve these paradoxes by addressing the plurality of identities in this community as stemming from its history as part of the Spanish Jewish diaspora and the Islamic lands beginning in 1391 as well as stemming from the community being integral to the French socialization process that began with the French conquest in 1830. This may explain why the Algerian Jewish community was the only one that the Israeli establishment chose to judge in a mock "public trial" initiated by the Jewish Agency (1963) for their low rate of emigration to Israel in contrast to the massive *aliyot* (waves of immigration to Israel) from the Islamic lands in the State's early years; this also explains why the scholarly community is unsure of whether to label the Algerian underground of World War II as French or Jewish.

In this article, I will employ Algerian Jewish education as a test case to clarify and concretize the fundamental insights briefly presented at the beginning of the paper.

## 2. French Colonialism and Judeo-French "Colonialism"

The French expeditionary force's invasion of Algeria in June 1830 marked the beginning of one hundred and thirty-two years of French rule over Algeria and essentially signaled the opening of the modern era of European colonialism. The French conquest was a bloody one. During this process, the rebellion led by Abd Al-Qādir was repressed, the rural population was dispossessed of its land, traditional society was destroyed, and the religious leadership was eliminated. Until 1870, a military government was tasked with keeping order among the diverse populations—Muslims, Berbers, Moors, Christians, and Jews. In 1870, France finally annexed Algeria and installed a civilian government.

It can be assumed that when Algeria's 16,000 Jews—residing in Algiers and its adjacent districts, Oran, and Constantine—saw the French armies arriving, they viewed the spectacle with concern. They remembered that in the not so distant past, the involvement of the European great powers had failed to successfully guarantee the Jewish community's safety or stability—I mean mainly the days of the Spanish conquests during the 16th and 17th centuries (Schaub 1999).

Algerian natives (Indigènes) quickly realized that the French had decided to rule the area with an iron fist, and, indeed, the French began making their mark on the

area. The French notion of colonialism unquestionably differed from that of the other European great powers. It is worthwhile noting the unique elements characterizing French colonialism and the important role that French education played in this process (Schwartzfuchs 1981, pp. 1–29).

**A Strong System of Government**: France made it clear to all concerned who was the ruler and who were the ruled, and it perpetuated clear disparities and inequality between the ruling colonialists and the indigenous society.

**The Aura of an Omnipotent State**: The French conquest and transformation of Algeria into a French colony gave France the image of being an enormously powerful state. This image was the result of an intentional division between the state and society, with the state's autonomy from society achieved by establishing clear criteria that advanced the growth of an elite and promoted its success. Thus, the ethos of the French bureaucrat who shows blind loyalty to the state was born—the French administration became a power unto itself and was not dependent upon society.

**Centralization:** Because of the French government's centralized nature—a function of the monarchical regime—the French sought to eliminate class distinctions and abolish groups with unique characteristics. Their aim was to transform society into one of individuals, pitting the power of the state against the individual rather than groups. This centralization was a prominent feature of all of France's state systems. Thus, the model of the Métropolitain (Metropolitan)—the mother state—was formed. The metropolitan, the center, determined the colony's development policy and rate change based upon the colonizing state's interests, not necessarily those of the colonial society. With the establishment of the Third French Republic (1870–1940), direct military rule over Algeria came to an end. From 1870, Algeria was an integral part of France administered by a civilian administration composed of branches of the various government ministries in Paris.

**Secularization**: France's nature as a secular state was central to its character. The French Revolution, while not necessarily anti-Christian, did end the Church's role as a locus of economic power, with all the Church's assets transferred into the state's coffers. The struggle between the Church and the state reached its apex with the enactment of the 1905 law separating religion and state. Thus, the colonial governance in Algeria was inherently secular although the local society was traditional.

**Disseminating Values through Education**: France considered education a highly effective tool in disseminating secularism and France's national values and in nurturing loyalty to France. Furthermore, it viewed education as crucial to progress and social mobility, to promoting the integration of the many ethnic groups within the local society, to shaping the immensely influential elites, and to creating the power structure of the state.

**Hierarchical Society**: Algerian colonial society was hierarchical, its strata determined by ethno-cultural criteria. On the top rung were members of the ruling French society; below it were those who had adapted themselves to French culture and received French citizenship; and on the bottom rung were those rejected by the ruling elites and/or those who were estranged from the upper stratum's values and distanced from these elites. Unlike the other colonial powers, France made it quite clear to its subjects, including the Jews, that anyone who adopted the ruling class's culture could enter the upper stratum.

**Imposing French Norms**: France's concept of colonialization was based upon the ambition to establish France's political, social, and cultural norms in Algeria. This went in tandem with a systematic effort to settle French citizens among the local population.

All these characteristics reveal the unique nature of the encounter between the French colonial power and indigenous society and portend the clash between them.

French Jewry played a decisive role in Algerian Jewry's process of Frenchification and in the radical transformations Algerian Jewry underwent. An investigation of French archives and the archives of French Jewry indicates that there was a high degree of correlation between the goals of the various bodies: an assimilatory goal promoted by the French regime and a desire to please the French establishment and its government by the

French Jews, who practiced a form of Judaism that was shaped by the Western Enlightenment movement.

Algerian Judaism was perceived by the French authorities as an integral part of the French colonial society. As a result, they viewed the continuum of Algerian Jewish history as a natural progression towards integration and assimilation into French colonial society, to the point where the Jews would lose their independent identity and uniqueness. Assimilationist forces within French Jewry help advance this trend toward integration, the most prominent of which was the Israelite Central Consistory of France—the new communal organization established by Napoleon Bonaparte. The Alliance Israélite Universelle—the supracommunal Jewish organization founded by the French-Jewish politician Adolphe Crémieux (1796–1880)—also played an important role.

French Jewry saw itself as the "Jewish people's firstborn." It further believed that it could help Jews throughout the world. That is, just as France had benefitted humanity by disseminating the values of the French Revolution, so, too, the Jews of France, who were the first Jews to obtain equal rights (1809)—and who, as Professor Michel Abitbol put it, imagined themselves to be "the tribe of Judah (Abitbol 1993)",—would benefit the world's Jews, no matter where they might be. This was particularly the case for those living in less advanced countries, such as the countries of the Spanish diaspora and Islamic lands.

Jewish solidarity, French patriotism, Orientalism, and a reformist ideology combined to fuel French Jewry's interactions with Oriental and Northern African Jews, in general, and with Algerian Jews in particular. The traditional Jewish sentiment of mutual responsibility only increased when French Jewry was granted emancipation. A sense of gratitude to France was transformed into absolute loyalty, blossoming into French patriotism. Orientalism and reform were two of the most prominent mindsets among Western European Jewry, especially among French Jewry. Indeed, they are two sides of the same coin. We may assume that the ever-increasing interest in the Orient among authors, artists, and French intellectuals—foremost among them the Romantic Alphonse de Lamartine (1790–1869)—that began at the dawn of the nineteenth century made an impression upon French Jews who were also extremely interested in the Jews of the Orient, North Africa, and the Land of Israel (Berchet 1985, pp. 3–20).

The Algerian Jewish community was unprepared for what was to come following the French conquest: social and spiritual crises spanning the late-eighteenth and early-nineteenth centuries and affecting primarily the Algiers Jewish community. As a result, many prominent sages left Algeria and the remaining Jewish judges and lay leadership—the wealthy *mukkadmim*—acted impiously and perverted justice. The decline of Algiers, the Jewish community's most important center, cast a dark shadow over the entire Algerian Jewish community. Furthermore, the mukkadmim, who were aggressive lay leaders with close connections to the Algerian rulers—the Deys and the Beys, in whom power accrued after the departure of the Ottoman Empire—were estranged from the Jewish community. They were primarily engaged in high politics, that is, in the reciprocal relationship between the local rulers and the "Sublime Porte" in Constantinople, and in international trade. They were members of a closed elite group that was composed of wealthy and esteemed Jewish families that were, however, not unified. Driven by their aspirations for individual financial success, they became caught up in the maelstrom of local politics. This elite—which functioned as patrons rather than leaders of the community—played a limited role in guiding the members of the community and preparing them for the challenges they would face in the modern era.

When the French arrived, the Jewish communities were primarily urban. Almost the entire Jewish population was concentrated in only ten communities. The four largest—Algiers, Oran, Tlemçen, and Constantine—contained 85% of the Jewish population of Algeria; the rest of the Jews lived in Béjaïa (Bougie), Médéa, Blida, Miliana, Mostaganem, Mascara, Annaba (Bône) and El Kala—near the Mediterranean coast—and Laghouat and Ghardaïa—located in the interior (the M'zab). After the French conquest, Algerian Jewry

continued to maintain its urban character, though they were divided into many medium-and small-sized communities.

### 3. The French Conquest's Influence on Changes in the Jewish Community

In the 1860s, significant headway was made on the civilian front: the Central Consistory in Paris placed Algerian Jewry's emancipation above all else. Renown figures and esteemed community members declared that Algerian Jews should be granted French citizenship. Leading this charge was Adolphe Crémieux (1796–1880), a Spanish Jew from the towns of Crémieux and Carpentras in Southern France. Born in the city of Nîmes in 1796 and trained in the law, Crémieux resolutely took upon himself the task of eliminating every instance of falsehood, injustice, and discrimination that the Jews suffered in France and elsewhere. It was no accident that he was among the members of the delegation of Western European Jews that came to Damascene Jewry's aid during the Damascus Blood Libel (1840). In addition, he was one of the founders of L'Alliance Israélite Universelle in 1860 and was appointed French Minister of Justice in 1870, a position that provided him with the rare opportunity of putting his principles into practice. As a result of his family drama—his wife Amélie Silny and his sons converted to Christianity in 1845—Crémieux was not allowed to take part in the Central Consistory of French Jewry. As a result, he had to channel his energies into serving Judaism and the Jews through an organization that at the time was still peripheral—L'Alliance Israélite Universelle—but which would in time take upon itself projects of great importance for the Jewish world.

Since the establishment of the Consistory in 1845, Algerian Jews had been required to abide by French law but were still officially considered indigenous peoples. Their legal status created confusion, especially in the area of family law: the courts' contradictory rulings and the legal confusion and lack of clarity led the French authorities to conclude that the only way to resolve this disarray was to grant Algerian Jews French citizenship (Rosenstock 1956; Szajkowski 1956).

Popular opinion among the Europeans in Algeria was shaped by republicans who were the "exiles of the Revolution of 1848", and who supported granting French citizenship to the Jews. As a result of the failure of the Revolution of 1848 and the establishment of the Second French Empire (1852), many republicans emigrated to Algeria, and they—like the liberals who arrived in Algeria during the period of the restoration of the monarchy—perceived Algerian Jewry's emancipation as a logical continuation of French Jewry's emancipation. They supported abolishing the military government that had ruled Algeria since 1830 and were convinced that to accomplish this, the French population had to be increased either by expanding French settlement efforts or by granting French citizenship to the Jews and other Europeans who were streaming into Algeria.

In contrast, the French military vociferously opposed changing the Jews' legal status, fearing that any such change would result in the abolition of the military regime due to the ruling powers' interest in protecting the governmental system from the natives. They also wanted to protect Algeria's economic resources from the Jews and to prevent a dangerous feeling of unrest among the Muslim population. The liberal approach and the approach opposing granting the Jews citizenship remained in constant tension until 1870.

The Algerian Jewish community's spokespeople—the consistory heads—declared that the community wanted French citizenship. They recruited the Central Consistory and public figures to help them in their battle. The edict that Napoleon III (1808–1873) issued after his visit in 1865—the Senatus Consulte Decree—failed to endorse the consistories' assumptions and did not prompt the influx of masses to get citizenship. According to this decree, all of Algeria's indigenous inhabitants—Muslims and Jews—could petition for and receive French citizenship on an individual basis as long as they declared that they were willing to renounce their personal status and waive their right to be judged by their traditional law. As mentioned, the response was minimal. Notwithstanding the public relations campaign and the consistories' vigorous attempt to persuade the people, only 3000 out of a populous of 23,000 Jews petitioned for citizenship. Algerian Jews did not want

French citizenship, viewing emancipation as a direct assault on Jewish religious identity and traditional society.

Adolphe Crémieux, over the course of his 17 trips to Algeria, built a strong relationship with Algerian Jewry and came to understand its character, and he therefore understood why it resisted French citizenship. He concluded that the Senatus Consulte had failed and that another approach had to be tried—granting citizenship to Algerian Jewry as a collective. Thus, as mentioned above, when Crémieux was appointed Minister of Justice in France's temporary government, he had the means to issue the 24 October 1870 decree applying French citizenship laws to the majority of Algerian Jewry, with the exception of Jews from the south, the Mzab. The "Crémieux Decree" made the Algerian Jews French citizens, with all the rights and obligations of French law now applying to them, including the obligation to be conscripted into the army and participate in elections. This decree was the first step in a process that led Algerian Jews to preside over community organizations and increase their involvement in colonial politics.

While the structural changes in communal organization and legal standing were forced upon Algerian Jewry, societal and economic changes resulting from their own free choices gradually emerged among Algerian Jewry. No doubt, the slogans of modernization and civilization provided a further impetus to the Algerian Jews into choosing these paths; however, these choices never eliminated the community's traditional dimension. Likewise, the secularization that Algerian Jewry later underwent was distinct from the secularization that the Western and Middle European Jewish communities underwent. Among Algerian Jews, tradition and modernity went hand-in-hand, and the secularization process was pragmatic, not ideological.

Universality, justice and equality, individual rights, the supremacy of reason and empirical knowledge over superstitions and prejudice, the expansion of human and intellectual horizons, the promotion of citizens based on their talents and abilities, personal hygiene and public sanitation, the rule of law and the abolition of subjective judgment (Chetrit 1990)—these were all messages that permeated the atmosphere the air and captivated their listeners, influencing Algerian Jews as they made their choices. Most Algerian Jews did, indeed, integrate into the main processes of modernization:

**Urbanization**: About 60% of Algerian Jewry lived in Algeria's three major cities—Algiers, Oran, and Constantine.

**Demographic Growth**: Within a period of about 120 years, the Algerian Jewish community grew eight-fold, from 16,000 to 130,000. This increase was the result not only of the Jewish population's fertility and the traditional emphasis on the Jewish family but also of improved medical care and sanitation and lower mortality rates.

**Geographical Dispersion**: The wealthy families began to move from the traditional Jewish quarter to the European neighborhoods.

**Employment Patterns**: As a rule, there was a decrease in "traditional professions", including crafts and trade, and, concurrently, an increase in the rate of wage earners; the number of women employed outside the family framework grew, and even a middle class was created of those working in the free professions or as bureaucrats.

**The Imprint of French Education**: The French school was the primary conduit of French culture. Its impact was not only educational and cultural but also mental and psychological. The school taught the French language while also transmitting values, norms, and an entire culture. The school gave birth to new patterns of identification derived from the French heritage. After the publication of the Crémieux Decree (1870) and the Jules Ferry Laws (1882), which mandated compulsory primary school education, most Algerian Jewish children received a French primary school education. This process was completed on the eve of World War One and illiteracy in the French language was unheard of in the northern parts of Algeria. The people understood that a French education was the means to improving their social status. The school took upon itself the burden of uniting or "Frenchifying" the population; it was the instrument deemed capable of unifying the

diverse ethnic groups. The Jews successfully adopted the French language and gradually abandoned their traditional Judeo-Arabic one.

These changes were highly significant. The Jews quickly differentiated themselves from the Muslim population and within two generations, they collectively became French "members of the Mosaic religion". It was not only the Crémieux Decree that made them French but also their psychological willingness to take the necessary steps to fulfill this dream, understanding the social advantages that their new status would bestow upon them. The social changes were gradual but undeniable—abandoning Judeo-Arabic for French, adopting French-European dress, and leaving the Jewish quarters for new neighborhoods, where a hitherto unknown relationship developed between the Algerian Jews and the Europeans and the French. The Algerian Jews gradually adopted French first names and came into more frequent contact with the French due to changes in society's professional and economic structure. The Jews became interested in the new ideas, values, and ideologies in vogue at the time, like secularization, nationalism, and scientific rationalism. French schools and French army service were the main forces pulling Jews into French society. The Jews quickly realized that the French school was a useful tool for advancing their upward social and cultural mobility. French education, both compulsory and free, quickly developed a positive image due to its clear advantages. Academic studies and French army service provided the Jews with the opportunity to encounter the "metropolis"—with all its real and illusory charms. Last but not least, military service heightened Algerian Jewry's patriotism, a value that gained expression during both world wars and in the political arena.

Indeed, from the end of the nineteenth century, Algerian Jewry experienced a steady decline in religious practice and in their connection to the synagogue, communal life, traditions, and religious customs, and in their respect for rabbinic authority. However, this process of acculturation was selective and accompanied by some reservations. Algerian Jews were, indeed, French, but they also preserved their distinctiveness, identifying with French culture but maintaining a certain distance. The Jews remained an ethnic community capable of preserving some of their unique characteristics within the colonial context, with the almost complete absence of intermarriage in Algeria signifying exactly how far Algerian Jews were willing to go. Algerian Jewry wished to assimilate the fundaments of French culture without being swallowed up by it.

Jewish education was another factor that enabled the Jewish community to tread the fine line between tradition and modernity for many years.

## 4. Jewish Education in Algeria—The Basis for Multiple Identities: The Consistorial Period (1830–1900)

Jewish education is the very bedrock of Jewish spiritual life and the observance of the commandments. Before the French conquest, Jewish education occupied a central role in the lives of Algerian Jewish communities, and this reality was made unequivocally clear by the uncompromising willingness adults demonstrated to toil and labor so the children could learn. This emphasis is reflected in the words of Rabbi Shimon Bar Tzemach Duran (the Rashbatz, one of Algeria's leading fifteenth-century Torah scholars). He declared that one must set aside part of the funds in one's estate (or the inheritance one leaves) to educate the orphans, that the community must commit significant funds to children's education, and that the status of those who teach young children be raised to be even comparable to that of the Torah scholars.

Education was fundamentally based on the study of sacred texts and its goal was to shape Jewish identity. The lessons usually took place in the synagogue or the house of study, and the children learned in a Talmud Torah or in a *Midrasch*. The adolescents and adults learned in the yeshiva, which also functioned as the place for teaching Halakha (legal rulings). The learning was connected to and conditioned upon everyday life. The halakhic debates were a function of the halakhic questions that arose from the daily lives and concerns of the community members, which were presented to the halakhic authority.

Similarly, the study of the written and the oral law were connected—as one unified corpus—to the weekly Torah reading (Aminoach 2004; Weinstein 1974).

The French Conquest in 1830 would later bring about changes to the face of Jewish education as the community faced new challenges and tests hidden in the fabric of the modern world. Until the establishment of the Consistory (1845), the *Midrashim* (pl. of *Midrash*) continued to be the main vehicle of Jewish education.

The St. Cloud Decree, issued on 5 November 1845, made the Consistory exclusively responsible for Jewish education on all levels and for designing a new pedagogical program for the *Midraschim.* This decree followed the Altaras Report, published on 1 November 1842, which would later be the basis and political platform for the founding of the Algerian Jewish Consistory, and it was the source for the changes that would take place in the field of Algerian Jewish education (Schwartzfuchs 1981, pp. 75–80). According to the St. Cloud Decree, the chief and regional rabbis' role included teaching the Jewish religion; repeating the obligation to obey French law and to be loyal to France and to protect it on every possible occasion; holding prayer services; praying for the peace of the king and his family in all the synagogues in their regions; supervising the Jewish daycare centers and schools; and overseeing the religious lessons given in them. The second half of the decree established directives clarifying all the technical and substantive elements of a Jewish education. In addition, the decree declared the establishment of daycare centers for children and Jewish schools for boys and girls, emphasizing that the public administrator would supervise them. The administrator was to consult with the Consistory regarding the hiring and firing of teachers, the code of discipline, the learning materials, and the establishment of school committees. The decree also established that "the instruction would include religious studies and the study of the French language (CAHJP, 1066)".

Thus, two types of Jewish educational institutions were created in Algeria—the revised and improved *Midrasch*, supervised by the Consistory, which continued to exist in its traditional form, and a Jewish school (Ecole Israélite), which the Consistory sought to establish in the image of the vision dictated by the Imperial Decree (1845). It was the Consistory's aspiration to make the Ecole Israélite model the leading one in the field of Jewish education.

Until the end of the 1860s and beginning of the 1870s, the two Jewish educational models—the traditional and the modern—could operate without any reciprocity between them. Had it not been for Crémieux's Decree (1870) and the Jules Ferry Law (1882), this dichotomy could have made its mark on Algerian Jewish education for generations (CAHJP, AL 1, 12.7.1852, p. 157).[1] However, this was not to be. The emancipation of Algerian Jewry and their subsequent obligation to follow the French law dictating compulsory primary school education made the notion of a Jewish school irrelevant. The Consistory's supervisory role over the *Midraschim* was expanded (1876), as they became the only vehicles for Algerian Jewish education. The Algerian Consistory's plans to found a Jewish high school also had no chance of coming to fruition, as the Central Consistory was entirely satisfied with the unmediated encounter with French culture that the Algerian Jews could find in the French government schools (Uhry, 1878, 1.1.1876).

The last two decades of the nineteenth century, therefore, were marked by an ongoing erosion in the status of Algerian Jewish education and by the Consistory's strong sense of dissatisfaction. During the 1890s, the Algerian Consistory underwent a soul-searching process and reorganized all areas of their responsibilities, including Jewish education. Powers within the community pushed for change, improvement, and transformation. In response to the severe wave of anti-Semitism that swept through Algeria in 1898, the community leaders believed that the community should regroup around the religious leadership and halt the process of assimilation and estrangement from Jewish heritage that was gaining momentum.

The French Consistory criticized how the *Midraschim* were being run in Algeria, feeling that these institutions were merely stagnating. The Central Consistory in Algeria warned the French Minister of Religions of this in written letters and reports: "Algerian

Jewish religion's state of affairs is getting worse, and it is necessary to correct the situation immediately by taking radical steps," states one of the reports. The report concludes, "The Consistory and the French regime need to urgently reach an agreement regarding the grave and shameful situation that everything concerning Algerian Jewry is caught up in—Jewish ritual, social support, and religious education" (CAHJP AL3, 10.11.1878, p. 159; see also CAHJP, AL4, pp. 47, 89, 95; CAHJP, AL5; CAHJP, HM 2 5106, January 1899, April 1900, February, December 1901, June 1902). The nineteenth century concluded with the Consistory's deep disappointment at the results that failed to reflect all the resources, efforts, and energy that had been invested since the Algerian Consistory had been established.

This crisis was the result of the Consistory's lack of autonomy and its constituents' perception of it as a tool of the regime intended to infuse the Jews with the French model of modernization and strengthen their loyalty to France. The Consistory, which had been imported from France, made the Jews feel like they had no internal leadership and that growth could come only from the intervention of external forces. It was only natural that a sense of estrangement would grow between the leadership and the community. The community's willingness to accept instruction from the leadership, which had not won their trust, was limited. The centrality of French education in the Consistorial worldview overshadowed the importance of Jewish education (Schwarzfuchs 1982).

At the end of the nineteenth century, the Consistory began to search for a partner—and, perhaps, for an organization that would take on sole responsibility—for the job of educating Algerian Jewry. The natural partner for such a venture was an organization that already had a close relationship with the French Consistory, an organization that more than any other emblazoned education on its banner—L'Alliance Israélite Universelle, which had been founded in 1860. The beginning of the twentieth century witnessed the transition from the Consistorial stage to the L'Alliance Israélite Universelle stage in Algerian Jewish education, while the local foundations for Jewish education were strengthened (Univers Israélites 50, 1898).

**5. The Alliance Enters the Field of Algerian Jewish Education (1900–1907)**

Albert Confino (1866–1958), the Alliance's official representative in Algeria from 1912 to 1955, explained the Alliance's reluctance to take on this communal role. According to him, the Alliance felt that it had nothing to contribute to Algerian Jewish children's education because as far as they were concerned, the Jews were already receiving the finest education possible—French education. However, the anti-Semitic unrest in Algerian (in 1898) convinced the Alliance's leaders in Paris that they had erred in their judgment regarding this community. Therefore, the Alliance decided that it should take upon itself the task of Algerian Jewish education on the Jews' behalf, so that they could help Algerian Jews complete the process of adopting French culture and, in so doing, indirectly combat Algerian anti-Semitism. The scholar Dr. David Cohen emphasizes that the Alliance was reluctant to use the same blueprint—the integration of Jewish and general studies—that they had employed throughout the Mediterranean basin for the Algerian "Jewish school" (Cohen 1995). Confino feared:

> Depriving our children of daily contact with their friends belonging to other faiths and religions, and, thus, slowing down the process of integration so devoutly to be wished for. This would have given our enemies at the time ammunition for calling out what they perceived to be our sectarianism. (Cohen 1995, p. 106)

Therefore, due to its desire to stand with the Algerian Jewish community in its time of strife and to prevent "social and cultural isolation," the Alliance proposed an original and novel solution: it would focus on teaching Judaism both through the Talmud Torah network and through charitable works, in the framework of the *Ha-Avodah* (the Labor) society, in which the students would also learn professions (CAHJP, HM 2 5190 a, b, 3.3.1903; Navon 1935, p. 126; Navon 1902–1903).

The leaders of the Alliance advocated for three fundamental principles that were included in the Algerian Talmud-Torah study program: education leading to an ethical life

"that is derived from the principles found in Tanakh"; knowledge of Jewish history designed to provide the students with a source of inspiration, experience, and identification; and Torah study as "the beginning point and origin of sacred studies, and this is the cornerstone of our entire educational enterprise." (Cohen 1995, pp. 112–13).

However, we must qualify these statements by clarifying exactly what the Alliance meant by "Torah," "Jewish religion," "Judaism," and "Jewish studies." The Alliance's attitude towards Jewish studies was predicated on loyalty to the values of the French Revolution and the appreciation of French Jewry's advancement under the banners of the Second French Republic and the Empire. From the Alliance's perspective, Judaism was primarily a religion. Every Jew was honor-bound to be faithful to the tradition. A Jew had to remain a Jew. However, there was no contradiction between loyalty to Judaism and loyalty to the values of the legacy of the French Revolution; in fact, the opposite was the case—the two were complementary. Thus, although the Alliance's leaders were avowedly secular Jews, they believed that Jewish studies had to be provided alongside the general ones prevalent in Western Europe at the time. Therefore, the Alliance supported "assimilation," in the most positive way the term was used at the time: adopting the most suitable path enabling the modern Jew to remain a Jew and simultaneously to be absorbed to the greatest degree in the national society in which he dwells.

The Alliance began operating its many activities with great enthusiasm and vigor, but it had to cope with the barriers to becoming integrated within the community. The first decade of the twentieth century was marked by the Alliance's struggle to enter the field of Algerian Jewish education. The Alliance's appearance in Algeria threatened the traditional educational system. It posed a cultural, economic, and social threat, and it created a struggle over student recruitment, the locus of authority, and the source of livelihood for rabbis and teachers. The Alliance's negative attitude toward the *Midraschim* only intensified the threat. When the Alliance, supported by the chief rabbis, decided to either abolish the *Midraschim* or merge them with their own framework, it signaled the opening of a prolonged struggle. Unlike the Consistory, which had supervised the *Midraschim* (1876), the Alliance sought to create a new kind of entity, albeit based on the local rabbi-teachers; unlike the Consistory, which wanted to establish a Jewish school based on the French blueprint alongside of the existing *Midraschim*, the Alliance aspired to unify the entire Jewish educational system.

In June 1907, the lengthy history of misunderstandings and disagreements between the Alliance, the Consistory, and the Jewish community came to a close. The heads of the *Midraschim* agreed to integrate the institutions under their aegis into the Alliance's framework. At this point, a Consistorial Educational Committee was appointed comprised of twelve members, among them the directors of the Alliance and the Algerian chief rabbis. This committee, which was responsible primarily for the financial side of Jewish education, demonstrated the Alliance's successful unification of the community's educational forces (CAHJP, HM 2, 5898 a, b; CAHJP, AL 7, p. 27, 30 October 1912, pp. 240–55, 250; CAHJP, AL 8, p. 72; Bulletin de l'AIU, Statistiques des Ecoles, 1911, pp. 645–49).

## 6. The Alliance and the Reform of the Algerian Jewish Education System (1907–1939)

While the first decade was devoted to conceptualizing the system and setting it up, the next two decades saw the educational system explore the question of education and cope with issues that surfaced.

**The rabbi teachers in Algeria** were primarily trained in the advanced yeshivas bearing the name *Etz Hayyim*. These particular institutions offered the highest level of Jewish education available in the three largest cities: Algiers, Oran, and Constantine. These yeshivas underwent ups and downs during the nineteenth century, as did the status of the rabbi teacher. With the approach of World War I, we witness an increasing awareness of the importance of the higher adolescent and adult education system and the profound importance of resurrecting Hebrew and Jewish studies on all levels. At the initiative of Moïses Scebat, the *Etz Hayyim* Society was founded, taking upon itself the task of raising the spiritual and educational level in Algiers (in 1920). The Alliance immediately joined

this movement. In 1928, *Etz Hayyim* societies with these same educational goals were established in Oran and Constantine. All these steps were taken against the backdrop of the strengthening of the rabbi teacher's status in Algeria. The rabbi teacher's status continued to strengthen until World War II. This was related to two processes. One was the local community's ever-increasing power in determining its own character and fate. This process had two significant consequences that influenced the rabbi teacher's status. The spiritual leadership that had been imported from France and that had, until this point, discriminated against the local leadership, gradually gave way to the local rabbis, thus leading to their increased stature. The Consistory also took care to ensure the rabbi teacher's stature and to undertake to find employment for the *Etz Hayyim* yeshiva graduates in their fields of specialization. The second process was related to the connection between the studies in the *Etz Hayyim* yeshiva and the general education, acquired in the French public school system. The rabbi teachers spent a substantial amount of time engaged in Jewish studies at the *Etz Hayyim* yeshiva, but gradually, an ever-increasing number of *Etz Hayyim* graduates earned the *Brevet Elémentaire* (granted at the end of Grade 9, signifying a basic French education), the *Certificat d'Etudes* (signifying the completion of France's core curriculum), and even the *Baccalauréat* (equivalent to an American high school diploma). Not only did the rabbi teacher's knowledge of the French language make communication with the students easier, but it also influenced the quality of their instruction. Furthermore, it was a positive sign of the rabbi teacher's social advancement and their acquisition of broader knowledge, which were also indicators of the rabbi teacher's enhanced status. The track for training rabbi teachers, which began in the *Etz Hayyim* yeshiva, reached the rabbinical seminary in Paris, gradually becoming a prestigious track attended by the very best students (CAHJP AL 3, p. 205; CAHJP AL 7, pp. 6, 198, 200, 233, 317; CAHJP AL 9, p. 10).

**Girls' Education:** Unlike the situation in France, where education for girls had been mandated since 1873, in Algeria, the girls were not part of any Jewish educational framework until the 1930s. Before the Alliance's arrival in Algeria, girls received no organized religious education. The Alliance wanted to introduce Jewish education for girls as soon as the organization was established in the educational arena—at the end of their first decade of activity in Algeria. As Alliance leadership wrote:

> The feminine element impacts on everything—on family life, on children's education, and on the community's future; we attribute the greatest possible importance to the religious and moral education of the girls who will become tomorrow's mothers in their own homes. (CAHJP HM 2 5898 a, b, 30 October 1912, pp. 25–26)

There was a gap between the girls' level of education in French and their knowledge of Judaism (Bashan 2006). The Alliance believed that the young and adult women gave equal weight to superstitions and legends as to Judaism's precepts. Confino stressed that:

> We attribute the greatest possible importance to the religious and moral education of the girls who will become tomorrow's mothers and in their homes will practice the Jewish tradition with devotion that is intended to make their children into good Jews and useful citizens for the motherland and the community. (CAHJP HM 2 5898 a, b, pp. 25–26).

The girls' education was not an immediate success, and it was only introduced in the large population centers—Oran, Constantine, and Algiers—beginning in 1930, 1935, and 1936, respectively. Parents in the smaller communities of the interior, who were more conservative in outlook, did not send their daughters to Talmud Torah, and only the boys attended. In the large communities, many women also refrained from sending their daughters to Talmud-Torah on Sundays and Thursdays, claiming that they were needed to perform housework.

**Moving beyond the Jewish Quarter Walls:** As the Jews moved beyond the walls of the traditional Jewish quarter, so did the boundaries of Jewish education expand towards the new neighborhoods. The decision to build synagogues with Talmud-Torah study halls next to them solved the problem created by the distance between these new neighborhoods and

the educational infrastructure, which initially remained solely located in the Jewish quarter. Eventually, a small distinction became apparent between the Jewish education offered in the traditional Jewish quarter and the new neighborhoods. The Jewish education in the Jewish quarter was perceived as more old-fashioned than that offered in the new neighborhoods, where the French language was more dominant and the pedagogical approach was more modern and open to changes (CAHJP, AL 9, p. 14; AL 10).

**The Reorganization of Higher Jewish Education—The *Etz Hayyim* Yeshivas.** As I have mentioned above, the *Etz Hayyim* yeshivas located in Algiers and Constantine were venerable communal institutions. Their influence and power were notable during the nineteenth century; however, at the beginning of the twentieth century, they underwent changes and a reorganization. The 1920s is marked as the period in which the enactment dictating "regulations and the design of an educational program suited to modern life" (private archive of Rabbi Maurice Zerbib, dedication of Yeshiva Etz Hayyim, Constantine, 1928; AL 7, p. 10; see also AL 7, pp. 198, 239, 317) was applied to these yeshivas in Algiers (in 1920) and in Oran and Constantine (both in 1928). The rebirth of the *Etz Hayyim* yeshivas in their modern form brought three issues back to the fore: the status of the rabbi teacher; the modernization of study, halakhic rulings, and pedagogy; and the connection between higher Jewish education and Jewish primary education. The founders of the revitalized yeshiva were certain about the yeshiva's centrality in the community's spiritual life and they maintained that the community could no longer turn a blind eye to whomever was appointed to run the yeshiva. They believed that the yeshivas of the past could not be compared to contemporary ones, as in the past, the yeshivas flourished, the rabbis were at the pinnacle of the social pyramid, and religious principles dictated the educational path taken by a child from a young age until entering the yeshiva. Furthermore, the yeshivas of the past, before the era of change, trained the rabbis, most—if not all—of whom were Torah scholars who received the Torah transmitted to them from generation to generation, without any changes being made in the methods of instruction. "That was a period in which all the Jews could proudly say "We are all Torah scholars. We all know the Torah."" (private archive of Rabbi Maurice Zerbib, dedication of Yeshiva Etz Hayyim, Constantine, 1928; AL 7, p. 10; see also AL 7, pp. 198, 239, 317). The crisis of faith that afflicted the generation and the lack of financial stability were—so they claimed—the stumbling blocks preventing the development of Jewish education on all its levels (CAHJP AL 7, p. 239).

The main educational and spiritual messages that the recharged and innovative—the renewed and renewing—yeshiva heads had to impart concerned the need to communicate with the student in a language that would penetrate his heart and that would interest him; the importance of addressing the student's intellect and not just focusing on what he could remember; the importance of illuminating the rationality of Jewish thought through the ages—accompanied by an appeal to the student's religious feelings; the importance of translating and explaining the texts being studied and making them intelligible to the student; the need to establish a synthesis between the study of Jewish culture—Jewish history and Jewish music—and the study of general culture—classical and modern French literature and French history—in a way that would allow Jewish culture to become a source of inspiration and identification; and most importantly, to present Jewish culture to the student in a clear, tangible, and harmonious fashion. By adopting this approach, the *Etz Hayyim* yeshiva integrated itself into the overall framework whose goal was to establish a new infrastructure that would support a fresh blueprint for Algerian Jewish education. Algerian Jewish education traditionally involved the study of the following books and subjects: Tanakh and Tanakh commentaries—with Rashi's commentary being the most basic and fundamental one; the Hebrew language and Torah cantillation notes; the *shurūḥ*—the Judeo-Arabic translation of the Holy Scriptures; Oral Torah and Jewish law; prayers; Kabbalah; Jewish thought; and the principles of faith (CAHJP HM 2 5898 a, b; CAHJP HM 2 5106; CAHJP HM 2 4948; Guedj 1887).

Some of these subjects were also taught in an integrative fashion. For example, Tanakh classes also incorporated the study of Hebrew, geography, ancient Near Eastern history,

prayer, and Jewish thought. The need for translations from Hebrew, Judeo-Arabic, and Aramaic into French created a dependency and connection between three central subjects in the Jewish educational system in Algeria—the Bible, Talmud, and Hebrew.

In that modern era, teaching Hebrew was in and of itself an innovation. The local rabbi teachers and the Alliance carefully considered new and more teaching approaches that were more effective than the older ones. Of course, after World War II, the various Land of Israel movements' influence on teaching modern Hebrew increased exponentially; however, the modern methodologies and the constant attempt to improve them were already a part of the educational system from the beginning of the twentieth century.

This was also true of Talmud study, which remained fundamentally traditional but was presented in a modern fashion, based on philosophical criteria reflecting Judaism's moral calculus as apprehended by the Sages of Antiquity. The crown jewel of this new curriculum derived from Jewish culture was the study of Jewish history. The Consistory and the Alliance encouraged the teaching of Jewish history, a field that was thought to shape the character of its students and that would help them cope with the "crisis of faith" afflicting the community by strengthening their faith and pride in their nation—they would learn the history of their nation, a nation scattered and oppressed throughout the ages, a nation that preserved its vitality and also continued to fulfill its divine mission.[2]

While the Alliance did promote the study of history, it also fought against the use of Judeo-Arabic in the Talmud Torah system. It believed that the use of Judeo-Arabic was in direct contradiction with Algerian Jewry's emancipation. Confino thought that "the task [of the rabbis] is to teach Hebrew and translate it into the country's official language. This had been done in the past during the Arab conquest, and, today, when France is in power, we should do it with French." (Cohen 1995, p. 122). This struggle exemplifies the lack of understanding that remained between the Alliance and the local Jews—the Alliance failed to understand Judeo-Arabic's singular position in the Algerian iteration of Judaism. It was not merely the lingua franca but a form of holy speech (*shurūḥ*) closely tied to the Holy Scriptures. The Alliance, deeply committed to the sacrosanct mission of spreading French culture, had difficulty respecting the local community's emotional needs and yearnings. The Alliance's battle was successful—on the eve of World War II, the French language obtained recognition in the Algerian Talmud Torah network. However, in traditional communities, such as Constantine, Judeo-Arabic did not disappear from the cultural landscape. Modern academic research has, indeed, revealed the centrality of Judeo-Arabic to Algerian Judaism even in later periods.

Dr. David Cohen reported that in the Alliance's "study house" there was a strong link between Jewish education and Jewish charitable enterprises. The *Le Travail* organization, founded in 1890, was its oldest venture. By World War II, the Alliance had founded thirty charitable enterprises throughout Algeria. Their programming included helping expectant women mothers and women who had given birth; issuing clothing to needy students; training seamstresses, embroiderers, and stenographers; aiding the orphans and the blind; providing dowries for brides; and the organization of a forum for Jewish culture that primarily planned lectures. We should note that the founders of these charitable enterprises were mainly graduates of Alliance-Algeria's Talmud Torah network. This indicates that there was an undeniable connection between the Alliance's socio-educational orientation and its production of leaders and community members who came to the aid of others (Cohen 1995).

## 7. Conclusions

The communal, social, and cultural changes that were accelerated by French colonialism altered Algerian Jewish education. However, the Spanish Jewish legacy provided the community with tools to cope with these changes. By the twentieth century, a reciprocal relationship was developed between the Alliance and the rabbi teachers that benefitted Algerian Jewish education. Slowly but surely, the local rabbi teachers developed a positive regard for the Alliance's plan to establish order in the Jewish educational system,

applying a systematic approach. Likewise, the Alliance finally realized that there was no need to negate the traditional foundations of Jewish education to bring about their desired educational reform. It was enough to simply repair the framework and add content to enable the community to advance towards Enlightenment culture, free of tensions or sensitivities. While some individual rabbis and Zionist leaders continued to disagree with the Alliance approach, they did not cast a pall over its fundamental activities, which as a rule successfully put down roots and were regarded positively throughout Algeria.

It is interesting to note that while the Alliance was a distinctly modernizing factor in most Mediterranean basin countries; in Algeria, it intentionally adopted a conservative stance: as in the past, Jewish education continued to mold Jewish identity and was not primarily functional as it was in other countries. The Alliance contended that in Algeria, the Jews received practical education in the French public schools, where the Jewish youth was exposed to French culture in all its aspects and, therefore, could achieve educational and professional success and integrate properly into French society. This was not only a necessary function of the Alliance's educational strategy but also a result of the local rabbi educators' significant influence, for while these men did integrate themselves into the Alliance educational system, they achieved such dominance within the system that its assimilatory characteristics disappeared without a trace.

The Alliance introduced equality into Jewish education; because of this blueprint for Jewish education, all the children in the community, not only those from families with the financial means, could benefit from a Jewish education. Furthermore, all signs indicate that the level of learning in the Talmud–Torah system was improved, due to students attending Talmud Torah year-round, every Monday and Tuesday, rather than attending only during the summer.

Jewish education in Algeria strengthened the Jews' loyalty to their Jewish identity and legacy and partially healed the fissures that increasingly appeared in the Jewish society exposed to an era of transformations. The multiplicity of identities that Algerian Jewry assumed during the French colonial period is the most important factor to recognize for accurately assessing Algerian Jewish history. The Algerian Jews' general and Jewish education both laid the foundations and provided the infrastructure for these multiple identities. Rabbi Y.L. Ashkenazi (Manitou), a member of the Parisian School of Jewish Thought, one of the architects of Hebrew identity, expertly analyzed the complexity of Algerian Jewish identity during the French colonial period in an interview I conducted with him. This complexity was both the basis for Algerian Jewry's rich and full Jewish identity and the Achille's heel of Algerian Jewish historiography:

The Algerian Jews did not consider the French colonial period to be a transition period, but rather the continuation of the exilic period with the addition of the French element. This identity derives from Hebrew-Arabic-Berber origins in terms of the general culture, possesses a Spanish character in terms of tradition and custom, a French character in terms of official citizenship, and a Hebrew influence from a liturgical standpoint, and all of this against the background of Andalusian music with all its nuances. There are Algerian Jewish figures who in their own unique ways succeeded in transforming their Jewish identity, which was fundamentally of a medieval Jewish-Arabic character, into a crypto-European, nineteenth and twentieth century one. Advancing and shepherding this "identity transformation" was no simple matter.

Our fathers' and grandfathers' generations were the ones who succeeded in doing this. As Israelis of French extraction and as a community indigenous to Algeria, we owe them our thanks. They were able to transmit a loyalty to Judaism that allowed us to once again become Hebrews after a lacuna of two-thousand years, during which we lived in parentheses. We think about them when we recite the verse praising faithfulness: "He is like a tree planted beside streams of water, which yields its fruit in season, whose foliage never fades, and whatever it produces thrives".

**Funding:** This research received no external funding.

**Data Availability Statement:** No new data were created or analyzed in this study. Data sharing is not applicable to this article.

**Conflicts of Interest:** The author declares no conflict of interest.

## Notes

[1]  "The French establishment criticized the inefficiency of the Jewish schools for the fact that they were not integrated into France's educational policy." (CAHJP AL 3, p. 205; see *Archives Israélites* 1856 (Constantine le 17.8.1856)).

[2]  See the exhaustive, 420-item inventory of the rabbinic literature written by Algerian rabbis that served, among other things, the Jewish educational system in the nineteenth and twentieth centuries (Charvit 2019, pp. 85–101).

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
