# Peer review of "Jewish Education in Algerian Jewish Communities—Multiple Identities in an Era of Change (1830–1939)"

_religions, doi:10.3390/rel15020163_

Round 1

Reviewer 1 Report

Comments and Suggestions for Authors

A fascinating and worthy contribtion to the study of Algerian Jewry after the French conquest. The author did an especially a superb job in presenting the controversies concerning Jewish Education in that time and place, as well as the tensions between local Algerian Jews and the French Consistory/AIU that essentially determined their fate.

line 40, 42, 59 etc - Author refers to something called "review for review," but it is unclear what this is.

line 44 The author is clearly trying to obviate an opinion or theory that maintains that Hebrew sources are somehow inadmissible in historical research, but cites no clear examples of scholars or works that touted such an approach. It is likely that researchers have included Hebrew sources simply because they do not read the language, rather than out of a deliberate effort to discount Hebrew sources as ahistorical or apologetic. This could be developed better, as the author's thesis seems to simply be saying that Hebrew sources can contribute to our understanidng of Algerian Jewry in the period in question, but that in and of itself is not novel if there is nobody who has ever stated otherwise. What are the "methodological limitations" of viewing Hebrew sources as legitimate? In general, the author could do a better job clarifying throughout the paper which insights and pieces of information come from Hebrew sources which come from other sources in order to better support its central thesis that Hebrew documents can be valuable for studying Algerian Jewish history.

line 650 the author might want to consider how teaching Hebrew as a subject matter reflects the community's position on Zionism.

I'm not sure why Rabbi Y.L. Ashkenazi (Manitou) is listed among the keywords if his presence is only acknowledged in the end of the paper, but he is otherwise irrelevant.

Comments on the Quality of English Language

This article demonstrates a commendable level of writing proficiency with minimal grammatical errors. The clarity of expression and coherence in presenting ideas greatly enhance its readability. The author's attention to language precision and structure significantly contributes to the overall quality of this work. 

Author Response

  1. please delete "review for review," its a mistake.
  2. On the contrary: I would like to emphasize that a serious historian should use all the sources available to him, while methodologically referring to the limitations and advantages of those sources.
  3. Indeed, the teaching of Hebrew in Talmud Torah was influenced at the beginning of the twentieth century by the modern Hebrew language in the Land of Israel
  4. Rabbi Y.L. Ashkenazi (Manitou) was a product of Jewish education in Algeria and his leadership and educational activity in France and Israel, so it is important to mention my research about him

Reviewer 2 Report

Comments and Suggestions for Authors

I read the article with great interest. The author has chosen a fascinating and important topic, and the article is very well written and logically organized. The author has also done a great job of finding useful archival sources to make their argument.

The article can use revision in the following areas. The article tends to focus on summary rather than argumentation and analysis, especially when listing out points using alphabetical letters. I think the argumentation needs to be more robust and direct throughout the article. Also, the author repeatedly cites "review for review." It seems clear that citations are needed here, and it's unclear why citations were not provided when the article was submitted.

See attached document for marginal notes.

Comments on the Quality of English Language

The quality of English is superb.

Author Response

  1. Line 95: I mean the Spanish and Portuguese conquest in the 18th century
  2. Please take into account the lector's comments regarding the punctuation and editing of the text (peer review)

Thank you

Reviewer 3 Report

Comments and Suggestions for Authors

This article is a strong addition to the literature. I recommend its swift acceptance after only a couple of very minor revisions. 

First is the ellipses on line 737-739. I don't understand why that sentence needs the ellipses. I recommend either removing them or making the entire sentence parenthetical. 

Second, lines 362-363 says that Jewish education's purpose was to "shape Jewish identity". I agree, but that's not its only purpose, although the sentence makes it sound that way. It would be more accurate to say, "among its many purposes included that of shaping Jewish identity." This acknowledges the salvific, linguistic and halakhic motivations for Jewish education, as well. 

Please watch for this kind of thing throughout the paper. 

Other than these small changes, I think you're good to go. Congratulations and well done. 

Author Response

  1. You can delete the ellipses
  2. Very agree. Please correct as the lecturer suggests: among its many purposes included that of shaping Jewish identity." This acknowledges the salvific, linguistic and halakhic motivations for Jewish education, as well.
  3. thank you